# Impact of a Mock OSCE on Student Confidence in Applying the Pharmacists’ Patient Care Process

**DOI:** 10.3390/pharmacy12020054

**Published:** 2024-03-24

**Authors:** Eleonso Cristobal, Kathryn Perkins, Connie Kang, Steven Chen

**Affiliations:** Alfred E. Mann School of Pharmacy and Pharmaceutical Sciences, Health Sciences Campus, University of Southern California, 1985 Zonal Ave, Los Angeles, CA 90089, USA; kscraig@usc.edu (K.P.); chens@usc.edu (S.C.)

**Keywords:** pharmacy, pharmacy education, OSCE, objective structured clinical examination, PPCP, pharmacists’ patient care process, simulated interviews

## Abstract

The Medical and Pharmacy Student Collaboration (MAPSC) student organization at the University of Southern California, Alfred E. Mann School of Pharmacy and Pharmaceutical Sciences, created an extracurricular, peer-led, virtual group mock objective structured clinical examination (MOSCE) to expose first-year pharmacy students (P1s) to the Pharmacists’ Patient Care Process (PPCP). The purpose of this study is to evaluate the impact of a MAPSC MOSCE on P1s self-reported confidence in applying the PPCP and on patient communication, medication knowledge, and clinical skills. An anonymous, optional, self-reported survey was administered to P1s before and after the event, where they rated their confidence on a scale of 0–100 (0 = not confident, 100 = certainly confident). The statistical analysis was a paired two-tailed t-test with a significance level of *p* < 0.05. A total of 152 P1s and 30 facilitators attended the MOSCE. One hundred thirty-nine students met the inclusion criteria and were included in the data analysis. There was a statistically significant difference in the change in self-reported confidence for all PPCP components and learning outcomes. The results of our study strongly indicate that introducing P1 students to the PPCP through a MAPSC MOSCE format is a valuable experience.

## 1. Introduction

In the United States, an objective structured clinical examination (OSCE) is becoming the standard method of non-experiential clinical assessment for Doctor of Pharmacy students [1]. During an assessment, students are evaluated by trained examiners as they perform various clinical tasks in an interactive, simulated setting, often involving a patient actor [2]. These tasks may include collecting a medication history or assessing the appropriateness of a medication for a patient. In pharmacy schools, OSCEs allow students to apply their clinical, communication, and decision-making skills, thereby assessing their confidence and competence in patient interactions.

In 2014, the Joint Commission of Pharmacy Practitioners created a patient-centered care model known as the Pharmacists’ Patient Care Process (PPCP) [3]. This initiative was created with the support of national pharmacy organizations and aims to ensure consistent and high-quality patient care across various practice settings. The PPCP comprises five key components: collect, assess, plan, implement, and follow-up/monitoring. Students must learn how to develop effective communication skills to collect relevant health information from patients. Utilizing their clinical training, students assess this information to create and implement a patient-centered care plan, including follow-up appointments and monitoring parameters [4]. The Accreditation Council for Pharmacy Education (ACPE), the national agency for the accreditation of professional pharmacy degrees, highlighted the importance of incorporating the PPCP and OSCEs into the curriculum as they are vital in a program’s accreditation process in their 2016 standards [5].

As a result of the COVID-19 pandemic, the pharmacy curriculum saw major changes relating to education infrastructure and delivery [6]. The abrupt shift to an online format necessitated the imminent and urgent development of a virtual OSCE [7,8]. At the University of Southern California (USC) Alfred E. Mann School of Pharmacy and Pharmaceutical Sciences, the Medical and Pharmacy Student Collaboration (MAPSC), a student-led organization, marshaled the development and implementation of a novel extracurricular, peer-led, virtual group Mock OSCE (MOSCE). This feat was aimed at filling a gap in hands-on clinical experiences left by the pandemic. The MOSCE format was used to simulate clinical experiences and familiarize first-year pharmacy students (P1s) with the PPCP. At this stage in the USC curriculum, students have received some exposure to the PPCP.

The MOSCE format was constructed in a peer-led format. Organizers recognized that peer-led teaching fosters collegiality and clinical application among students in a controlled environment [9,10]. Despite limited literature, there is evidence to suggest a virtual OSCE format is well-received by students, can offer an experience as engaging as in-person, and may be an efficient alternative method for examining students’ competencies [11,12,13]. To the authors knowledge, there are no studies assessing the impact of a PharmD student-led extracurricular virtual clinical experience, such as a MAPSC MOSCE, on P1s self-reported confidence in applying the PPCP. The purpose of this study is to describe the implementation of the extracurricular, peer-led, virtual group MOSCE, and to assess its impact on P1s self-reported confidence in applying the PPCP and on patient communication, medication knowledge, and clinical skills.

## 2. Materials and Methods

### 2.1. Study Context

Student board members in the MAPSC organization conceived, developed, and executed the entire MOSCE program in response to the COVID-19 pandemic. This involved creating a peer-led virtual group design, tailoring a patient case for P1 learning objectives, recruiting facilitators and students, marketing the event, and administering surveys for research purposes.

The virtual group MOSCE format was intentionally designed to optimize the learning experience for participating students while simultaneously broadening access to the event. This virtual format also facilitated increased involvement of student and resident volunteers in teaching roles, requiring minimal faculty work hours as an extracurricular activity.

Faculty advisors on this study verified the patient case accuracy and provided quality assurance for survey questions, research objectives, and the MOSCE’s applicability to P1 students’ learning objectives.

In 2021, MAPSC hosted two pilot MOSCEs. The first and second pilot events hosted 53 and 128 P1 students, with 11 and 119 fully completed pre- and post-surveys, respectively. Learnings from these events laid the groundwork for this study, including the timing of the administration of surveys, the formatting of questions, and standardizing mock patients and preceptors as one facilitator. This study was conducted for the third MOSCE, hosted in the spring of 2022. P1s were recruited from the PHRD 516 Non-Prescription Therapies course. At the time, the investigators were student board members of the MAPSC organization, and one faculty author was the course coordinator of PHRD 516. The USC Institutional Review Board determined this study to be exempt.

### 2.2. Mock OSCE Design and Implementation

The MOSCE was conducted over 1.5 h via Zoom using approximately 30 breakout rooms. Each breakout room consisted of a team of five P1s and one facilitator (faculty, resident, fourth-year student, or third-year student) who played dual roles as both a patient and preceptor (see Figure 1). A master document in Microsoft Excel, shared with all participants one week prior to the MOSCE, included a schedule, student team and role assignments, and hyperlinks to individual team documents. Each team document included identical patient cases, question guides, and blank recommendation sheets, enabling teams to collaboratively collect, assess, plan, and implement a treatment regimen.

The patient case used in the MOSCE was crafted by MAPSC student clinic directors, purposely created to align with P1 learning objectives for their Non-Prescription Therapies course. The case was then verified by the course coordinator of PHRD 516. The patient case incorporated three chief complaints (pain, fever, and cough), a history of present illness, past medical, family, and social history, along with an admission medication list. In the case design, each treatment regimen had to include pharmacological and non-pharmacological recommendations, while taking into consideration social determinants of health, to comprehensively address the various health and medication-related issues discovered during the MOSCE. 

#### 2.2.1. P1 Students’ Roles

MAPSC randomly assigned P1s on each team to either a SCHOLAR or MAC role in a 3:2 ratio, indicating the type of patient interview they would conduct based on the QuEST SCHOLAR-MAC process [14]. Illustrated in Figure 1, this process utilizes the SCHOLAR and MAC acronyms as essential pharmacy interviewing techniques, specifically for the “collect” phase of the PPCP in a patient consultation. The QuEST process was used to structure the “assess, plan, and implement” phases of the PPCP. The objective of the QuEST SCHOLAR-MAC process was to evaluate a patient’s condition quickly and thoroughly, provide guidance on their self-care needs, and make recommendations for appropriate OTC and non-OTC treatments.

SCHOLAR students focused on collecting information related to a patient’s chief complaint. The simulated patient presented with three chief complaints, with one SCHOLAR student managing each complaint. On the other hand, MAC students were responsible for collecting an accurate and up-to-date medication list, with medication history questions distributed evenly between two students.

Prior to the MOSCE, students only had access to the MOSCE schedule. No information was provided by the MAPSC coordinators on how to apply the QuEST SCHOLAR-MAC process and its subsequent interviewing techniques except in the “Introduction” portion of the MOSCE. The expectation was for students to utilize their knowledge of QuEST SCHOLAR-MAC and medication reconciliation skills gained from previous classes, work experience, and PHRD 516.

#### 2.2.2. Facilitator’s Role

MAPSC hosted an optional 30 min preparation session for facilitators. For those who could not attend, a recorded training video was provided. Facilitators assumed the role of a patient during the “collect” and “implement” phases of the interview (see Figure 1) and transitioned to the role of preceptors during the “assess” and “plan” phases. The “follow-up” phase of the PPCP was omitted since this patient encounter was designed as a one-time event. Facilitators received a completed team document, serving as a finished key for the patient case, clinical recommendations, and blank team documents. This document included patient-response transcripts for the facilitator patient role and a completed treatment plan for their preceptor role. The intention was to allow facilitators to switch between patient and preceptor roles seamlessly to ensure each team received identical information in their “collect, assess, plan, and implement” phases of their interviews and treatment plans.

### 2.3. Self-Reported Survey Design and Administration

The survey was developed by the student authors of this study and verified by the faculty advisor. Questions were adapted from validated instruments, including the PPCP Self-Efficacy Survey [15] and the Four Habits Coding Scheme [16], to align with the objectives and information conveyed in the MOSCE (see Appendix A). Thirty questions were divided among three learning outcomes: patient communication (11 questions), medication knowledge (7), and clinical skills (12). Clinical skills questions were further split into PPCP components: collect (3), assess (5), plan (3), and implement (1). The three learning outcomes were chosen as they are considered essential to contemporary pharmacy practice per 2016 ACPE standards [5]. For each question, P1 students rated their self-reported confidence on a scale of 0–100 (0 = not confident can do; 50 = moderately can do; 100 = certainly confident can do). The self-reported confidence scale of 0–100 was based on Bandura’s theory of self-efficacy, which states that when one assesses their perceived ability to carry out certain actions, a continuous scale should be used rather than interval scales such as Likert-scales, as people tend to avoid the extreme positions, which may distort the results [17]. The same optional survey was administered before and after the event.

### 2.4. Data Collection and Analysis

To be included in the study, each P1 student had to complete the pre- and post-surveys within 24 h before and after participation in the MOSCE. The surveys were administered to students individually through Qualtrics. The data were directly exported to Microsoft Excel, and any identifiable information, such as emails, was promptly deleted by the investigators. The primary outcomes were changes in self-reported confidence in (1) applying the PPCP and (2) patient communication, medication knowledge, and clinical skills. The statistical analysis was a paired two-tailed t-test with a significance level of *p* < 0.05. Outliers were removed from the sample population by grouping together scores for all PPCP components and learning outcomes, using the interquartile range method to normalize the data. A subgroup analysis evaluated differences in baseline characteristic questions in (1) gender groups, (2) age groups, (3) previous experience, and (4) those who’ve previously attended a MOSCE versus those who had not (see Appendix A). The statistical analysis for subgroups was a t-test with unequal variance. Inclusion criteria were P1s who attended the MOSCE, achieved 100% completion status for both surveys, and consented to participate in the study (see Figure 2). Participation in the MOSCE and survey was voluntary, but attendees were awarded 1% extra credit to their PHRD 516 course grade from the course coordinator.

## 3. Results

A total of 152 P1s and 30 facilitators (faculty (1 person), residents (5), fourth-year students (2), and third-year students (22)) participated in the MOSCE. One hundred thirty-nine (91%) P1 students met the inclusion criteria and were included in the data analysis. As described in Table 1, the study population included 78% females (*n* = 108). More than 70% of students had less than 1 month of previous medication counseling or experience collecting medication histories. Ninety-two (66%) students previously participated in telehealth training, or MOSCE.

### 3.1. PPCP Outcomes

The change in self-reported confidence was significant for all PPCP components (*p* < 0.0001; see Table 2). The highest pre-scores were found in the collect and assess phases compared to later PPCP components. The greatest mean change in self-reported confidence was found in the plan and implement sections, which had the lowest pre-scores of 49 and 45, but had the greatest mean change of 28 and 31, respectively (see Figure 3). The removal of the outliers did not change the outcomes of any statistical analysis.

### 3.2. Learning Outcomes

Change in self-reported confidence was significant in all learning outcomes (*p* < 0.0001; see Table 2). Patient communication had the highest pre-scores of 64, with the lowest mean change of 18 points. The greatest mean change was found in learning outcomes with the lowest pre-scores (see Figure 4); clinical skills had the lowest pre-scores of 51 but the highest mean change of 24 points. The removal of the outliers did not change the outcomes of any statistical analysis.

### 3.3. Subgroup Analysis

Students who previously attended a MAPSC MOSCE had slightly higher pre-confidence scores in all outcomes. Conversely, students who did not attend a MOSCE had higher changes in mean scores in all outcomes (see Table 3). When comparing students who “had” vs. “had not” previously attended a MOSCE, the only statistically significant difference was the mean change in self-reported confidence in the “plan” (25 vs. 33, *p* = 0.04) and “implement” (28 vs. 36, *p* = 0.02) phases of the PPCP. There was no difference in outcomes across different age groups or with previous telehealth or medication history experience.

In the subgroup analysis comparing gender groups in the PPCP components, there was no difference in pre-, post-, or change in mean self-confidence scores (*p* > 0.05; see Table 4). When comparing the learning outcomes, there was only a significant difference in mean pre-scores for medication knowledge (female 50, male 59; *p* = 0.02) and post-scores (female 73, male 80; *p* = 0.01). The relative mean change between the pre- and post-scores between genders was not significant (*p* > 0.05) The removal of the outliers did not change the outcomes of any statistical analysis.

## 4. Discussion

Pharmacists are the only healthcare professionals who lack federal recognition as healthcare providers, despite scores of rigorous studies demonstrating that pharmacist clinical services consistently improve patient outcomes and medication adherence while reducing the total cost of healthcare [18]. The traditional role of pharmacists commonly recognized by the public involves the dispensing and compounding of medications. However, pharmacists’ clinical training and patient care roles have expanded, allowing them to prevent and manage chronic diseases. The PPCP provides the profession with a standardized approach to managing medication therapy for patients and optimizing health outcomes [3]. As a result, it is essential to teach the PPCP early in pharmacy education to better prepare students for the evolving role of pharmacists and advanced pharmacy practice opportunities.

At the USC Mann School of Pharmacy, the initial focus of collecting patient information, facilitated by tools like QuEST SCHOLAR-MAC, occurs in the first half of the spring semester within the P1 curriculum. However, a more comprehensive understanding of all PPCP components, such as assess, plan, implement, and follow-up, is not introduced until the PHRD 520 Introduction to Therapeutics course is taught in the second half of the spring semester. The MOSCE took place during PHRD 516, but prior to PHRD 520.

The timing of the MOSCE in the P1 curriculum likely contributed to higher pre-scores in the “collect” and patient communication sections compared to the other PPCP components or learning outcomes. By the spring of their first year, students had been taught how to effectively gather patient-related health and medication information. During PHRD 516, students are introduced to processes for assessing patient-related information to create and implement a treatment and care plan. This stage of the curriculum marks the initial skill development around the latter components of the PPCP (assess, plan, implement, follow-up/monitor). In addition, during this stage, students begin developing their knowledge about the most common drugs and are introduced to concepts surrounding the application of clinical skills. Pharmacy students do not begin properly applying the latter components and respective learning outcomes until their therapeutics courses in the second-year (P2) and third-year (P3) of the USC Mann Curriculum. This portrays itself in the pre-scores of collect, assess, plan, and implement; the latter components, which have the least exposure, have the lowest pre-scores relative to the earlier components, such as “collect”. Similarly, with patient communication, medication knowledge, and clinical skills, there is a progressive decrease in pre-scores as one moves towards sections that are developed later in P2/P3 years (medication knowledge and clinical skills). When comparing the mean change, the PPCP components (assess, plan, implement) and learning outcomes (medication knowledge, clinical skills), all of which are emphasized in the subsequent P2/P3 years of the USC Mann curriculum, had the greatest improvement in self-reported confidence. The mean pre-scores for all measured outcomes were roughly 50, which meant students perceived themselves as moderately capable of carrying out the PPCP or learning outcomes. The post-scores were all roughly 20 points or greater, excluding patient communication. These results showed students perceived themselves as previously being moderately capable to now being above average capable of carrying out various components of the PPCP and the learning outcomes as a result of attending the MOSCE. These results suggest the MOSCE was effective in exposing students to all PPCP components, regardless of their previous exposure.

In a subgroup analysis comparing students with prior MOSCE or telehealth experience to those without, a significant difference emerged in the “plan” and “implement” phases of the PPCP. Students who had attended a previous MOSCE had already encountered the unique format where facilitators changed roles between patient and preceptor for these phases. This prior exposure may have played a role in the observed increase in self-reported confidence among students who’ve previously attended. In addition, previous exposure to working on MAPSC MOSCE team documents, including creating, planning, and implementing a treatment plan, may have contributed to an increase in students’ confidence in the “plan” and “implement” phases. Interestingly, the lack of significance in the “collect” and “assess’’ components suggests, irrespective of MOSCE attendance, that students, in general, are comfortable communicating with their fellow students (see Table 3). This corroborates with the study authors’ presuppositions about P1 students’ skill levels in relation to their curriculum. For these early PPCP components, this comfort level had no discernible impact on whether students attended a MOSCE or not.

Several studies have elucidated the effectiveness of faculty-led pharmacy capstone courses in enhancing students’ confidence in applying the PPCP [19,20,21,22]. Phillips et al. describe a P3 capstone course designed to prepare students for fourth-year rotations. Assessment of P3 students (*n* = 134) involved weekly quizzes and two practical examinations, and the authors found a statistically significant increase in mean grades and confidence in applying the PPCP between the two exams [19]. Noureldin et al. assessed self-efficacy in applying three PPCP components (collect, assess, and plan) for P3 students after completing a capstone course [20]. Results indicated a statistically significant increase in self-efficacy for all components. At the University of Buffalo, Maerten-Rivera et al. engaged P2 pharmacy and physician assistant students in an interprofessional activity over a semester, observing a significant increase in confidence in applying the PPCP [21]. Meanwhile, Rivkin described an introductory pharmacy capstone course to expose P1 students to the PPCP through faculty-led simulation interviews [22]. Evaluation included grading students on their ability to recognize the PPCP during an in-class assignment, with mean examination scores of 83.7%, and over 86% of students believed they progressed in their understanding of the PPCP. In the aforementioned studies, students demonstrated increased confidence and competency from the beginning to the end of the semester, with faculty-led courses serving as the primary teaching mechanism. Each study had a distinct focus: Philips and Noureldin focused on P3s during an in-person class, Maerten-Rivera focused on P2s and was the only virtual format, and Rivkin focused on P1s during an in-person class. None of these studies explored the impact of a solely extracurricular, peer-led, virtual format on P1 students’ self-reported confidence in applying the PPCP. The MAPSC MOSCE serves as an illustrative example of how pharmacy students can organize and execute events to improve confidence in applying their clinical and patient communication skills. Regardless of the format, the benefits of simulated patient interviews are well-demonstrated in the literature [1,11,12,13,19,20,21,22,23].

Student-led pharmacy organizations are a common extracurricular activity at the USC Mann School of Pharmacy, where students pursue board positions to develop both their pharmacy and leadership skills. Per 2016 ACPE standards, the fostering of organizations within schools is important in the accreditation process [5]. The COVID-19 pandemic highlighted how adaptable organizations can be, particularly through the development and implementation of valuable co-curricular events. The MOSCE, a significant initiative, was predominantly developed and coordinated by student leaders of MAPSC, with guidance from a few faculty mentors. The concept of peer-to-peer teaching was first proposed in 1976 by Ann Bragg and further developed in 1988 by Whitman and Fife; it suggests that barriers broken down in a peer-teaching model enhance socialization and allow students to gain the necessary information without feeling the undue stress from external environments [24]. The MOSCE design, as an extracurricular, peer-led event, intentionally mitigated the high-stakes environment commonly associated with OSCEs. For example, Raier-Lorimer et al. described the implementation of a student-led mock OSCE in a medical school. After the event, students reported a lack of stress, a sense of empowerment, and an increase in confidence in working with patients [10]. The MOSCE data infer a similar theme, suggesting a peer-led model may be a feasible approach to improving P1 students’ confidence in communicating with patients and applying the PPCP.

Student-led organizations, like MAPSC, may play a pivotal role in addressing gaps in curriculum and may provide a resource to overcome barriers associated with OSCEs. Common concerns about implementing OSCEs in curricula include the cost of standardized patients and building space, faculty workloads, and the difficulty of incorporating OSCEs into an existing curriculum [25,26]. For example, the average cost of OSCEs was $13 to $25 per hour per standardized patient [25], and an estimated 8.2 person-hours per student would be required to develop and implement an OSCE [26]. At USC Mann, with each PharmD cohort being under 200 students, implementing OSCEs has clear cost and time considerations. The MAPSC MOSCE offers an alternative solution that is affordable and sustainable. The MAPSC MOSCE was executed with zero finances and minimal faculty time and workload. While the MOSCE format should not replace curriculum learning, it serves to identify areas needing reinforcement or review and fosters communication between student-led organizations and faculty. Results from the MOSCE can inform PHRD 516 and other P1 course coordinators about tailoring curricular activities towards the PPCP and other hands-on learning experiences.

Lastly, telehealth has steadily influenced health care delivery and utilization in the past two decades [8]. Jamil et al. conducted a tele-OSCE during the COVID-19 pandemic, with 94% of the students (*n* = 17) finding the activity operationally easy to undertake and not detering their exam performance. Comparing final grades to a face-to-face OSCE from the previous year, there was no statistically significant difference in exam scores [11]. Similarly, Grover et al. described a virtual OSCE for medical students in the UK, resulting in a significant improvement in self-reported confidence across domains like medical history taking, communication, and data analysis (*p* < 0.0001) [13]. Both participants and faculty found the virtual OSCE to be as engaging and interactive as in-person experiences. While limited literature suggests virtual OSCEs as a viable alternative for teaching pharmacy practice skills and telehealth communication, this study alone does not conclude that MOSCEs replicate in-person OSCE experiences [8,12]. Mock examinations, however, may aid in preparing students for annual competencies and serve as a preparatory clinical experience during the P1 year.

This study has several limitations. First, the study’s scope focused on self-reported confidence and perceived ability to apply the PPCP as a result of attending an extracurricular MOSCE. However, self-reported confidence does not give meaningful insights into a student’s true competency. Next, there may be two types of scoring biases in the PPCP components: (1) a neutral scoring bias where students may have perceived themselves as “average” and scored accordingly, thus potentially explaining the similarity in post-confidence scores [27]; and (2) a social desirability [28] and confirmation bias where students want to see themselves as more competent so that they may be viewed more favorably by others, thus ranking higher scores [29,30]. The absence of standardized patients or preceptors introduced variability in the facilitators’ knowledge levels and abilities, making it challenging to ensure a consistent experience for each participating P1. This could have influenced confidence scores. Lastly, the lack of information about whether students who completed the pre- and post-surveys also attended the MOSCE poses a limitation. Given that the data were anonymous, attendance cannot be verified, potentially distorting the results.

Future directions of this study will be to determine the impact of the MOSCE on students’ competencies in their graded OSCE courses within the P1 curriculum.

## 5. Conclusions

The results of our study strongly indicate that introducing P1 students to the PPCP through a MAPSC MOSCE format is a valuable experience. The MOSCE, a large-scale, extracurricular, peer-led, virtual event, brought together 182 individuals from the USC Mann School of Pharmacy. Given the challenges and gaps in U.S. healthcare quality and safety, contemporary pharmacy practice emphasizes the delivery of specialized and high-quality care. Our findings suggest that early exposure to the PPCP through an extracurricular, simulated patient interview increases P1 students’ confidence in applying fundamental pharmacy skills. Further research is needed to determine the relationship between P1 students’ self-reported confidence as a result of attending a MAPSC MOSCE and their ability to appropriately apply clinical knowledge and skills through their exams and clinical rotations.

## Figures and Tables

**Figure 1 pharmacy-12-00054-f001:**
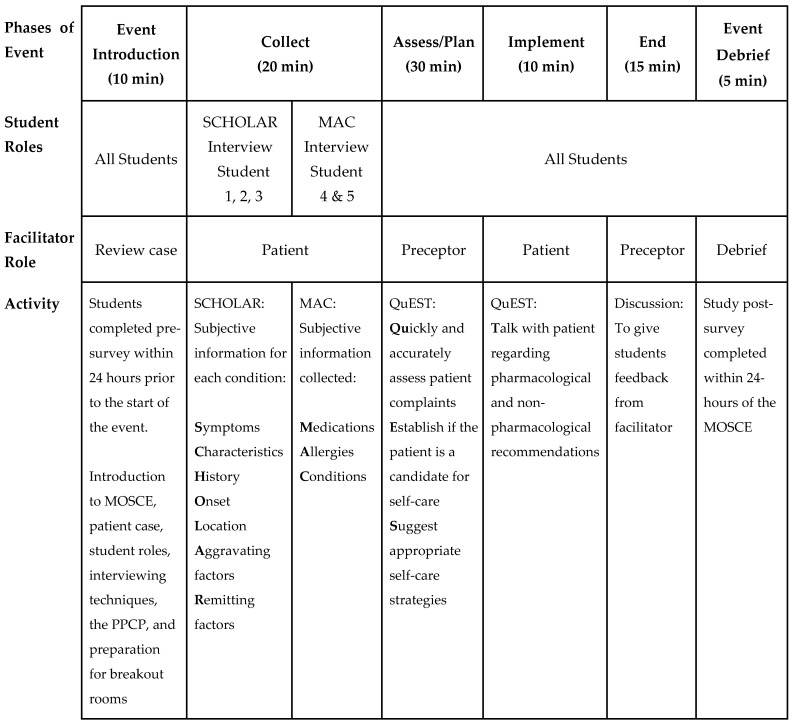
Composition of individual breakout rooms, student and facilitator roles, and activities for 1.5 h MOSCE.

**Figure 2 pharmacy-12-00054-f002:**
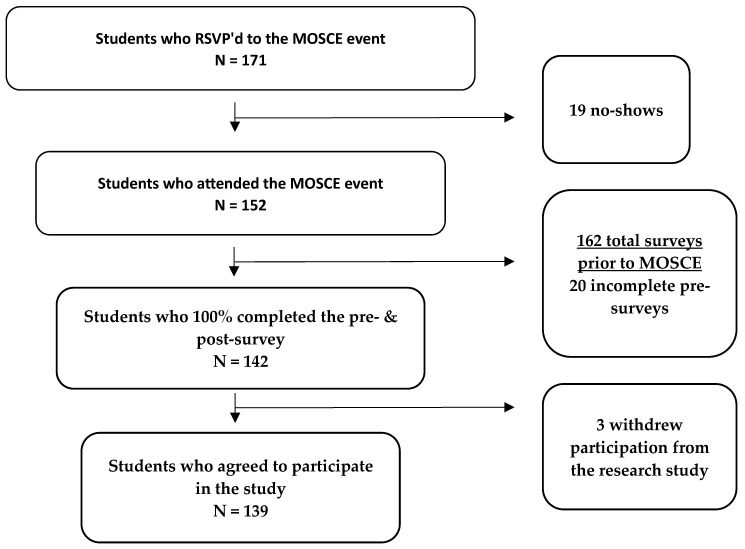
Study population (N = 139).

**Figure 3 pharmacy-12-00054-f003:**
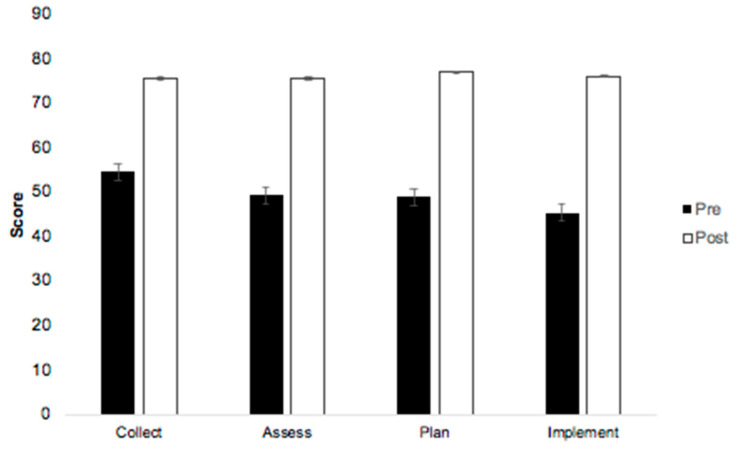
Changes in student self-reported confidence in applying the PPCP components pre- vs. post-MOSCE.

**Figure 4 pharmacy-12-00054-f004:**
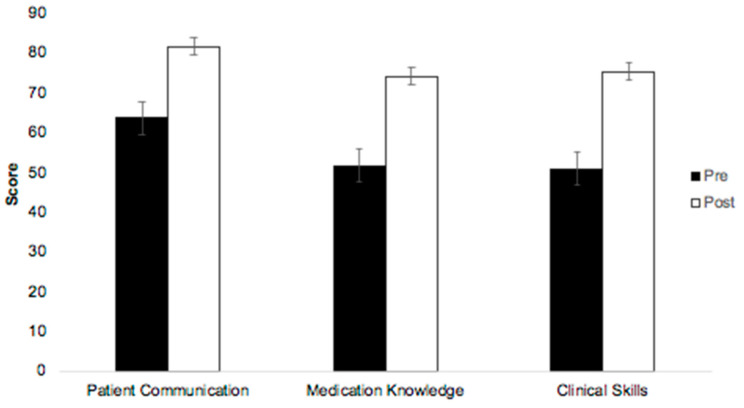
Changes in student self-reported confidence in the three learning outcomes pre- vs. post-MOSCE.

**Table 1 pharmacy-12-00054-t001:** P1 students baseline characteristics include gender, age, previous year of experience in counseling, and experience in collecting medication histories.

Study Participant Baseline Characteristics	
**Gender**	*n* (%)
Male	31 (22)
Female	108 (78)
**Age**	
18–24	99 (71)
25–34	36 (26)
35–44	3 (2)
45–54	1 (1)
**Previous years of experience counseling patients on medications**	
<1 month	98 (71)
1–6 months	32 (23)
6 months–1 year	6 (4)
>1 year	3 (2)
**Previous years of experience collecting medication histories**	
<1 month	102 (73)
1–6 months	25 (18)
6 months–1 year	7 (5)
>1 year	5 (4)
**Previously attended a telehealth clinic or MOSCE**	
Yes	92 (66)
No	47 (34)

**Table 2 pharmacy-12-00054-t002:** Average student self-reported confidence scores for pre- and post-MOSCE for PPCP components and learning outcomes.

Outcome (Number of Survey Questions)	Self-Reported Confidence Scores				*p*-Value
	Pre	Post	Change	*n*	
**Collect (3)**	55	76	21	126	*p* < 0.0001
**Assess (5)**	49	76	27	131	
**Plan (3)**	49	77	28	132	
**Implement (1)**	45	76	31	132	
**Clinical Skills (12)**	51	75	24	131	
**Medication Knowledge (7)**	52	74	22	132	
**Patient Communication (11)**	64	82	18	130	

**Table 3 pharmacy-12-00054-t003:** Subgroup analysis of change in self-reported confidence in students who attended a previous MOSCE versus those who have not.

Outcome	Prior MOSCE Attendance			No Prior MOSCE Attendance			*p*-Values (Significance of *p* < 0.05)		
	Pre	Post	Change (*n*)	Pre	Post	Change (*n*)	Pre	Post	Change (*n*)
**Collect**	55	75	20 (84)	53	76	23 (42)	0.46	0.65	0.14 (126)
**Assess**	51	75	24 (89)	46	77	31 (42)	0.25	0.49	0.06 (131)
**Plan**	51	76	25 (88)	45	78	33 (44)	0.16	0.56	**0.04 (132)**
**Implement**	48	76	28 (88)	41	77	36 (44)	0.10	0.70	**0.02 (132)**
**Clinical Skills**	52	75	23 (87)	49	76	27 (44)	0.30	0.67	0.09 (131)
**Medication** **Knowledge**	52	74	22 (90)	51	76	25 (42)	0.85	0.52	0.35 (132)
**Patient** **Communication**	65	82	17 (86)	61	82	21 (44)	0.32	0.80	0.09 (130)

**Table 4 pharmacy-12-00054-t004:** Subgroup analysis of change in self-reported confidence between different genders.

Outcome	Female			Male			*p*-Values (Significance of *p* < 0.05)		
	Pre	Post	Change (*n*)	Pre	Post	Change (*n*)	Pre	Post	Change (*n*)
**Collect**	54	74	20 (95)	56	79	23 (31)	0.66	0.18	0.41 (126)
**Assess**	48	75	27 (102)	52	78	26 (29)	0.36	0.36	0.79 (131)
**Plan**	48	77	29 (103)	53	77	24 (29)	0.26	0.99	0.18 (132)
**Implement**	44	76	32 (103)	49	78	29 (29)	0.31	0.63	0.40 (132)
**Clinical Skills**	50	75	25 (101)	54	78	24 (30)	0.38	0.29	0.92 (131)
**Medication** **Knowledge**	50	73	23 (103)	59	80	21 (29)	**0.02**	**0.01**	0.56 (132)
**Patient** **Communication**	64	82	18 (101)	62	82	20 (29)	0.52	0.82	0.25 (130)

## Data Availability

The datasets presented in this article are not readily available because the data are a part of an ongoing study. Requests to access the datasets should be directed to Eleonso Cristobal.

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
