# Peer review of "Impact of a Mock OSCE on Student Confidence in Applying the Pharmacists’ Patient Care Process"

_pharmacy, 2024, doi:10.3390/pharmacy12020054_

Round 1

Reviewer 1 Report

Comments and Suggestions for Authors

1.  Please be consistent with reference to COVID-19.  There are several places where it is referenced to both "COVID" instead of "COVID-19".

2.  Based on the design, this is best described as a peer-led, virtual "group" MOSCE as 5 P1 students were in each breakout room with 1 facilitator, 3 collected data using SCHOLAR, 2 using MAC and all 5 were involved in the assess/plan, implement and discussion. 

3.  It would be of interest to your readers how the MOSCE impact the annual competencies, the actual OSCE.  How was OSCE performance after MOSCE compared to the year prior to the implementation of MOSCE?  It is likely beyond the scope of this study/manuscript, perhaps comment on it.

Otherwise, great work in the midst of pandemic!

Author Response

  1. Updated to "COVID-19".
  2. I agree. We've added "group" where it is appropriate. To line 9, 66, 74, and 77.
  3. Added future directions to lines 425-426. This paper for MOSCE's impact on competencies in graded courses is my next paper I'm writing.

Thank you for your time.

Reviewer 2 Report

Comments and Suggestions for Authors

Thank you for the opportunity to review this paper describing a student org-led mOSCE and how it impacted student confidence about the PPCP. Overall, there are minor spelling/grammar issues in multiple places, but otherwise, the article is written well. I have some suggestions that would make the paper better detailed below.

Introduction, lines 42-44: Seems like the California piece isn’t needed given that you’re wanting to publish in an international journal (and you include the ACPE information right afterwards)

Introduction, lines 48-55: I’m unclear why the pandemic itself left a gap in familiarizing students with the PPCP. Was it included in in-person OSCEs but not virtual ones? Or was it not taught at all? (in which case, the pandemic information could be removed). The “why” of the study needs to be clearer.

Materials and Methods: I know you cite Bandura’s self-efficacy theory, but I still question the use of such a large rating scale. Was there any anchoring done for the students besides 0, 50, and 100 (i.e. what is the difference between a 50 and a 60?)? And what is the difference between ratings close together, like 52 vs 54?

Discussion: You found a statistically significant difference overall, but what difference would have been educationally meaningful? This needs to be clear. Ideally, this would have been used for a power calculation, but it doesn’t look like that was done (if it was, please include)

Discussion: There’s also the limitation of social desirability, people want to be seen as more confident after participating in an optional activity.

Comments on the Quality of English Language

Minor spelling/grammar issues, but otherwise fine.

Author Response

  1. Introduction, lines 42-44: Seems like the California piece isn’t needed given that you’re wanting to publish in an international journal (and you include the ACPE information right afterwards) Response: removed from article.
  2. I’m unclear why the pandemic itself left a gap in familiarizing students with the PPCP. Was it included in in-person OSCEs but not virtual ones? Or was it not taught at all? (in which case, the pandemic information could be removed). The “why” of the study needs to be clearer. Response: the MOSCE was created due to the drastic reduction in "hands-on clinical experiences." We filled a gap since students were remote and had limited hands-on clinical experiences where students could apply their pharmacy skills. We created the MOSCE so students could learn how to apply the PPCP in a simulated situation. Students had some exposure to PPCP prior to the MOSCE. Please see lines 53-57.
  3. Materials and Methods: I know you cite Bandura’s self-efficacy theory, but I still question the use of such a large rating scale. Was there any anchoring done for the students besides 0, 50, and 100 (i.e. what is the difference between a 50 and a 60?)? And what is the difference between ratings close together, like 52 vs 54? Response: Bandura's self-efficacy continuous rating scale is used throughout the literature and was chosen instead of a Likert scale. The ratings are based on a 0, 50, and 100 score rating and are clearly defined. The differences between a 52 and a 54 in the context of "moderately can do" is negligible. A 60 versus a 50 would be slightly higher than "moderately can do" but again, this is negligible. The question posed is to determine how does one know when someone has improved? Due to the nature of self-efficacy and perception, to determine if something was significant it would be as a result of the p-value. A more objective form would be based on a grade; however, that was not the scope of this study. Please see line 166.
  4. Discussion: You found a statistically significant difference overall, but what difference would have been educationally meaningful? This needs to be clear. Ideally, this would have been used for a power calculation, but it doesn’t look like that was done (if it was, please include) Response: the study's scope was to describe the implementation of the MOSCE and evaluate change in confidence pre- and post-event. This question refers to my 2nd manuscript, which will follow this paper, and has to do with confidence affecting competence. This question does not go with the aims of this study. Please see lines 425-426.
  5. Discussion: There’s also the limitation of social desirability, people want to be seen as more confident after participating in an optional activity. Response: please see lines 415. 

Thank you again for your comments and strengthening the overall aim of this paper.

Reviewer 3 Report

Comments and Suggestions for Authors

The authors did a nice job of clearly explaining exactly what went into delivering this complex activity! In addition to the targeted suggestions below, I think there are some sections that get a little wordy and can be tightened up for a quicker read. Otherwise, I enjoyed reading this and think that this does a nice job of creating a road map for similar activities at other schools. 

Intro line 54: suggest wording instead "this feat was aimed AT FILLING a gap..."

Intro line 57: suggest removing "In addition" - instead "Despite limited literature, there...."

Intro line 65: was the study actually able to evaluate the effects on patient communication, medication knowledge and clinical skills? Or did it evaluate students' perceptions of the their own abilities in the aforementioned? Without a score it seems like a stretch to say that it evaluated those things. 

Materials and Methods, line 85: "Learnings from these events laid groundwork for this study INCLUDING timing...and standardizing MOCK patients and preceptors as one facilitator"

2.2.1 PI Students' roles, line 118: 3:2 ratio because you say SCHOLAR before MAC and 3 students did scholar?

2.2.1 PI Students' roles, line 132: instead of master document, maybe just say schedule?

2.2.1 PI Students' roles, line 132: what do you mean no information was provided about QuEST SCHOLAR MAC? It then goes on to say that students should utilize their knowledge of it from previous classes. It may be more clear to spell out what background students received on the QuEST SCHOLAR MAC process anywhere, then specify that no reminders, or no additional information was provided at this session? (also what was given in the pre-event portion?) 

Line 322: What do you mean by students being comfortable communicating with peers? Peers as in patients? Or peers as in the P2s, P3s, residents, etc. playing the role? Or each other? 

Line 344: "Philips and Noureldin on P3 students, Maerten-Rivera on P2 students in a virtual format, and Riven on P1 students" (not parallel structure currently) 

Author Response

The authors did a nice job of clearly explaining exactly what went into delivering this complex activity! In addition to the targeted suggestions below, I think there are some sections that get a little wordy and can be tightened up for a quicker read. Otherwise, I enjoyed reading this and think that this does a nice job of creating a road map for similar activities at other schools. Response: thank you. the paper goes in length to describe the MOSCE implementation and results. The methodology can get lengthy but we believe it's important to describe the whole thought process around the MOSCE.

Intro line 54: suggest wording instead "this feat was aimed AT FILLING a gap..." Fixed

Intro line 57: suggest removing "In addition" - instead "Despite limited literature, there...." Fixed

Intro line 65: was the study actually able to evaluate the effects on patient communication, medication knowledge and clinical skills? Or did it evaluate students' perceptions of the their own abilities in the aforementioned? Without a score it seems like a stretch to say that it evaluated those things. Response: fixed. We evaluated the change in self-reported confidence. We removed "evaluate effects" to mitigate any confusion.  Please see lines 67-69.

Materials and Methods, line 85: "Learnings from these events laid groundwork for this study INCLUDING timing...and standardizing MOCK patients and preceptors as one facilitator" Fixed

2.2.1 PI Students' roles, line 118: 3:2 ratio because you say SCHOLAR before MAC and 3 students did scholar? Fixed

2.2.1 PI Students' roles, line 132: instead of master document, maybe just say schedule? Fixed

2.2.1 PI Students' roles, line 132: what do you mean no information was provided about QuEST SCHOLAR MAC? It then goes on to say that students should utilize their knowledge of it from previous classes. It may be more clear to spell out what background students received on the QuEST SCHOLAR MAC process anywhere, then specify that no reminders, or no additional information was provided at this session? (also what was given in the pre-event portion?) Response: Please see lines 134-139. We made it more clear about students QuEST SCHOLAR MAC knowledge and the information provided for the MOSCE.

Line 322: What do you mean by students being comfortable communicating with peers? Peers as in patients? Or peers as in the P2s, P3s, residents, etc. playing the role? Or each other? Response: please see line 330. We changed peers to "fellow students".

Line 344: "Philips and Noureldin on P3 students, Maerten-Rivera on P2 students in a virtual format, and Riven on P1 students" (not parallel structure currently) Fixed

Thank you again for your time and consideration. We appreciate your feedback to improve the overall aims of this study.